# Effect of Zinc Oxide Incorporation on the Antibacterial, Physicochemical, and Mechanical Properties of Pit and Fissure Sealants

**DOI:** 10.3390/polym15030529

**Published:** 2023-01-19

**Authors:** Ji-Won Choi, Song-Yi Yang

**Affiliations:** 1Department and Research Institute of Dental Biomaterials and Bioengineering, Yonsei University College of Dentistry, Seoul 03722, Republic of Korea; 2Department of Dental Hygiene, Konyang University, Daejeon 35365, Republic of Korea

**Keywords:** pit and fissure sealant, zinc oxide, antibacterial effect, physicochemical property, mechanical property

## Abstract

This study aimed to evaluate the antibacterial, physicochemical, and mechanical properties of pit and fissure sealants containing different weight percentages of zinc oxide nanoparticles (ZnO NPs). The following amounts of ZnO NPs were added to a commercially available pit and fissure sealant (BeautiSealant, Shofu, Japan) to prepare the experimental materials: 0 wt.% (commercial control (CC)), 0.5 wt.% (ZnO 0.5), 1 wt.% (ZnO 1.0), 2 wt.% (ZnO 2.0), and 4 wt.% (ZnO 4.0). The antibacterial effect against *S. mutans* was confirmed by counting the colony-forming units (CFUs) and observing live/dead bacteria. In addition, ion release, depth of cure, water sorption and solubility, and flexural strength tests were conducted. When compared with the CC, the experimental groups containing ZnO NPs showed zinc ion emission and significantly different CFUs (*p* < 0.05) with fewer live bacteria. ZnO NP addition reduced the depth of cure and water solubility and increased water sorption in comparison with the CC (*p* < 0.05). However, all groups showed similar flexural strength (*p* > 0.05). The pit and fissure sealants containing ZnO NPs exhibited antibacterial activity against *S. mutans* with no negative effects on physicochemical and mechanical properties, and thus, these sealants can be ideal secondary caries prevention material.

## 1. Introduction

One of the most common chronic infectious disorders worldwide is dental [1]. The presence of endogenous cariogenic bacteria such as *Streptococcus mutans* (*S. mutans*) and sugar or fermentable carbohydrate exposure to the tooth surface are the main causative factors for dental caries [2]. Due to the shape and depth of the occlusal surface, dental caries most commonly occurs in occlusal pits and fissures in young adults and children. In particular, occlusal surface caries in the first and second molars accounts for approximately 90% of all cases of dental caries [3]. The morphological complexity of the occlusal surface, including the presence of deep pits and narrow fissures that trap food and microorganisms, contributes to the progression of occlusal caries. Caries prevention is essential because caries can cause problems with speech, appearance, and psychology, as well as abnormal tongue habits and masticatory defects [4]. Therefore, the use of pit and fissure sealants, which can both treat and prevent early enamel caries, is widely accepted as a popular preventive therapy for dental caries [5].

The pit and fissure sealant acts as a physical barrier, preventing the accumulation of bacterial microorganisms and food debris to protect the teeth from destruction [6]. Since the development of the first pit and fissure sealant using methyl cyanoacrylate by E. I. Cueto in 1966, various products have been used as pit and fissure sealants. Glass ionomer-based sealant (GI), which can release fluoride ions, was introduced in 1974 by J. D Mclean and A. D Wilson. The components of GI not only act as a physical barrier but also help to remineralize the tooth structure, restoring its strength and function [7].

Currently, composite resins containing bioactive fillers based on surface pre-reacted glass (S-PRG) technology, which can continuously release multiple ions such as fluoride (F^−^), sodium (Na^+^), strontium (Sr^2+^), silicate (SiO_3_^2−^), borate (BO_3_^3−^), and aluminum (Al^3+^) ions, are being used as pit and fissure sealants. The mechanism underlying S-PRG technology is known to involve an acid–base reaction between fluoroboraluminosilicate glass and polycarboxylic acid, resulting in material biofunctionality [8]. Previous research has demonstrated that dental products containing S-PRG fillers can prevent tooth enamel demineralization [9] and provide acid-buffering ability [10] because of the constant release of these ions. Some studies have also suggested that the multiple ions released by S-PRG fillers may have antibacterial properties, since decreased bacterial adhesion and less biofilm accumulation were observed on resin composites with S-PRG fillers [11,12]. However, other studies have reported that the concentrations of ions released by the S-PRG filler is insufficient to inhibit bacterial growth [13,14]. In addition, the S-PRG-filler-based resin composite did not inhibit demineralized interfacial biofilm penetration [11].

Since the pit and fissure sealants’ antibacterial properties are important, attempts have been made to improve their antibacterial activity by incorporating antibacterial materials [4,15]. In particular, the previous studies have demonstrated that pit and fissure sealants containing nanosized particles as fillers such as silver nanoparticles provide significantly better performance as pit and fissure sealants than materials containing non-nanosized fillers [16,17,18]. Zinc oxide nanoparticles (ZnO NPs) are antibacterial materials with superior antibacterial properties to silver nanoparticles by producing reactive oxygen species [19,20,21]. Furthermore, the biocompatibility to human gingival fibroblast cells and L929 mouse fibroblast cells of ZnO NPs in dental and medical applications has been confirmed, and ZnO NPs have been used in composite resins, endodontic sealers, and adhesive cements in dental fields [22,23,24,25].

Thus, the purpose of this study was to develop an S-PRG technology-based pit and fissure sealant containing ZnO NPs and evaluate its antibacterial, physicochemical, and mechanical properties. The null hypothesis was that the antibacterial, physicochemical, and mechanical properties would not differ between pit and fissure sealants with and without ZnO NPs.

## 2. Materials and Methods

### 2.1. Preparation of Experimental Materials

A commercially available S-PRG technology-based pit and fissure sealant (BeautiSealant, Shofu, Kyoto, Japan) was used according to the manufacturer’s instructions. To prepare the experimental material, ZnO NPs (particle size, <100 nm; Sigma-Aldrich, St. Louis, MO, USA) were prepared. To investigate characterizations of ZnO NPs, particle size analyses were conducted using a particle size analyzer (Mastersizer 2000, Malvern, Worcestershire, UK). To confirm the shape of ZnO NPs, observation using scanning electron microscopy (SEM; JEOL-7800F, JEOL Ltd., Tokyo, Japan) was performed. The ZnO NPs were then mixed homogeneously with the pit and fissure sealant at the following weight concentrations using a high-speed mixer (Speed mixer; Hauschild, Hamm, Germany): 0.5 wt.% (ZnO 0.5), 1.0 wt.% (ZnO 1.0), 2.0 wt.% (ZnO 2.0), and 4.0 wt.% (ZnO 4.0). In addition, a pit and fissure sealant without ZnO NPs was used as the commercial control (CC) (Table 1).

### 2.2. Evaluation of Antibacterial Effects

#### 2.2.1. Bacterial Culture and Growth Conditions

*Streptococcus mutans* (*S. mutans*; No. 5365; Korean Collection for Type Culture, Jeollabuk-do, Korea) was used to evaluate the antibacterial effects of the experimental materials. *S. mutans* was cultured in brain heart infusion (BHI; Difco, Sparks, MD, USA) broth at 37 ± 1 °C for 48 h. Using a spectrophotometer (ELISA reader; Epoch, BioTek, Winooski, VT, USA), the optical density (OD) of the bacterial suspension at 600 nm was adjusted within the range of 0.4–0.6 by adding BHI broth.

#### 2.2.2. Morphological Assessment of the Bacteria Attached to the Specimens

Disc-shaped molds (diameter, 10 mm; thickness, 1.0 mm) were prepared, and after mixing the pit and fissure sealant with the ZnO NPs concentration described in Section 2.1, the experimental materials were filled in the molds and covered with a polyester film. To remove excess material, a slide glass was used to apply pressure to the specimen. Each experimental material was irradiated for 20 s on each side. The cured specimens were removed from the mold and sterilized using ethylene oxide gas. Sterilized specimens were placed on a 12-well cell culture plate (SPL Life Science, Gyeonggi-do, Korea). One hundred microliters of bacterial suspension (OD: 0.4–0.6 at 600 nm) was spread onto each specimen surface and then incubated at 37 ± 1 °C for 48 h. The specimens were gently washed twice with phosphate-buffered saline (PBS; Welgene, Gyeongsangbuk-do, Korea), fixed for 24 h in Karnovsky’s fixative solution, washed for 30 min in 0.1 M phosphate buffer, and post-fixed with O_s_O_4_ for 2 h. The specimens were then dehydrated with a series of graded ethanol solutions (50–100%) before being dried for 2 h using a critical point dryer (Leica EM CPD300; Leica, Wien, Austria). The specimens were then sputtered with platinum using an ion sputter (Leica EM ACE 600; Leica, Wien Austria) and observed using a field-emission scanning electron microscope (FE-SEM; Merlin; Carl Zeiss, Oberkochen, Germany) at 15.00 kV.

#### 2.2.3. Assessment of the Antibacterial Effect by Counting Colony-Forming Units

Disc-shaped specimens (10 mm × 1 mm) were fabricated using the method described in Section 2.2.2. A bacterial suspension (100 μL) was applied to each specimen surface and incubated at 37 ± 1 °C for 48 h. The specimens were washed twice with PBS. To detach bacteria from the specimen surface, the specimens were immersed in 1 mL of BHI broth and sonicated for 10 min. After collection of bacteria from the specimens, the bacterial suspension was diluted, spread onto BHI agar plates, and incubated at 37 ± 1 °C for 48 h. The antibacterial rate was calculated by counting colony-forming units (CFUs) using the obtained values in the following equation: antibacterial rate (%) = (1 − (CFUs of remaining bacteria on the specimen surface/CFUs of attached bacteria on the specimen surface)) × 100. The experiments were performed in triplicate.

#### 2.2.4. Evaluation of Bacterial Viability

To confirm the viability of the adherent bacteria, the specimens were fabricated as described in Section 2.2.2. Bacterial suspensions (100 μL) were applied to the surface of the specimen and incubated at 37 ± 1 °C for 48 h. Each specimen with attached bacteria was stained using a LIVE/DEAD staining kit (LIVE/DEAD BacLight Viability Kit; Thermo Fisher Scientific, Waltham, MA, USA) according to the manufacturer’s protocols. The stained specimens were observed under a confocal laser microscope (LSM880; Carl Zeiss, NY, USA). SYTO9 was used to indicate green fluorescence for live bacteria, and propidium iodide stain was used to indicate red fluorescence for dead bacteria.

### 2.3. Analysis of pH Variations

Bar-shaped molds were prepared with dimensions of 25 mm × 2 mm × 2 mm. The experimental materials were filled into the molds and covered with a polyester film. To remove the excess material, pressure was applied to the specimen with a slide glass. The experimental materials were then cured by overlapping the irradiation regions (20 s each) from both sides after mixing each concentration of pit and fissure sealant and ZnO NPs as described in Section 2.1. The cured specimens were removed from the mold and stored in distilled water (DW) at 0.14 cm^3^/1 mL for 1, 2, 4, 7, 14, 21, 30, 60, and 90 days. The DW was replaced at each time point, and the pH of the storage solution was measured after removing the specimens from DW at each time point. The pH measurements were obtained using a digital pH meter calibrated to pH 4.01, 7.0, and 10.01 (Orion 4 Star, Thermo Fisher Scientific, Waltham, MA, USA). The measurements were repeated thrice.

### 2.4. Evaluation of Zinc Ion Release

Bar-shaped specimens (25 mm × 2 mm × 2 mm) were fabricated using the same method described in Section 2.3. The polymerized specimens were removed from the mold and stored for 1, 2, 4, 7, 14, 21, 30, 60, and 90 d at 0.14 cm^3^/1 mL of DW, which was replaced at each time point. After removing the specimens from DW at each time point, the amount of ion emission was measured using an inductively coupled plasma optical emission spectrometer (ICP-OES; Optima 8300; PerkinElmer, Waltham, MA, USA). The measurements were repeated six times.

### 2.5. Assessment of the Depth of Cure

The depth of cure of the experimental materials was determined in accordance with ISO 6874 (2015) [26]. Each experimental material (*n* = 6) was fabricated in a cylindrical stainless-steel mold 6 mm in length and 4 mm in diameter. A polyester film was applied to the top, and excess material was removed using a slide glass. The specimens were vertically irradiated for 20 s. After polymerization, the uncured material at the bottom of the specimen was gently removed using a plastic spatula. A digital Vernier caliper (Mitutoyo Co., Kawasaki, Kanagawa, Japan) with an accuracy of 0.01 mm was used to measure the height of the polymerized material. The measurements were repeated three times and the average and standard deviation values were calculated.

### 2.6. Water Sorption and Solubility

The water sorption and solubility of the experimental materials were determined in accordance with ISO 4049 (2019) [27]. Six polymerized disk-shaped specimens with a diameter of 15.0 ± 0.1 mm and height of 1.0 ± 0.1 mm were prepared from each group. The diameter and height of the specimens were measured with an accuracy of 0.01 mm to obtain the volume (*V*), and the constant mass (m_1_) was obtained to an accuracy of 0.1 mg using a digital balance (XS105; Mettler Toldedo AG, Greifensee, Switzerland). The specimens were then individually placed in a 6-well plate (Cell Culture Plate; SPL Life Sciences Co. Ltd., Pocheon, Korea) filled with 10 mL of DW and stored at 37 ± 1 °C. After 7 d, the disk specimen was removed from DW, the surface water was blotted away, and the weight (m_2_) was measured 1 min after removing it from the DW. The specimens were then placed in a desiccator and weighed daily until a constant mass (m_3_) was obtained. Water sorption and solubility were calculated using Equations (1) and (2), respectively, and the measurements were repeated six times.
(1)$$W_{sp}\left(µg/{\mathrm{mm}}^{3}\right)=\frac{\left(m_{2}-m_{3}\right)}{V}$$
(2)$$W_{sl}\left(µg/{\mathrm{mm}}^{3}\right)=\frac{\left(m_{1}-m_{3}\right)}{V}$$

### 2.7. Three-Point Flexural Strength Assessment

The three-point flexural strength of the pit and fissure sealants was confirmed in accordance with ISO 4049 (2019) [27]. Six bar-shaped specimens (25 mm × 2 mm × 2 mm) were fabricated using the method described in Section 2.3. All the specimens were stored at 37 ± 1 °C in DW for 24 h prior to the flexural strength test. The specimens were loaded to fracture by three-point bending using a universal testing machine (Model 5942; Instron, Norwood, USA) with a span length of 20 mm and crosshead speed of 1 mm/min. The flexural strength (σ) was calculated using Equation (3).
(3)$$\mathrm{Flexural}\;\mathrm{strength}\;\left(\sigma \right)=\frac{3Fl}{2bh^{2}}$$
where *F* is the maximum load exerted (*n*), *l* is the distance between the supports (mm), *b* is the specimen width, and *h* is the height (mm). The measurements were repeated six times.

### 2.8. Statistical Analysis

The antibacterial effect, pH variations, zinc ion release, depth of cure, water sorption and solubility, and flexural strength of the experimental pit and fissure sealants were analyzed using one-way ANOVA (SPSS 25, IBM Co., Armonk, NY, USA), followed by Tukey’s statistical test. Statistical significance was set at *p* < 0.05.

## 3. Results

### 3.1. Characterization of ZnO NPs

The particle size distributions of ZnO NPs are presented in Figure 1. The average diameter was 1575.30 ± 218.70 nm. The morphology images of the ZnO NPs are shown in Figure 2. The shape of ZnO NPs appeared to be nanorods with soft aggregation.

### 3.2. Antibacterial Effects

#### 3.2.1. Morphology of Attached Bacteria on the Specimen

Figure 3 shows that *S. mutans* aggregated into long chains of diplococci was attached to the specimen surface. Surface attachment of *S. mutans* was more dense on the CC than in the other groups. *S. mutans* adhesion, on the other hand, decreased as the amount of ZnO NPs increased. ZnO 4.0 had a lower distribution of attached *S. mutans*.

#### 3.2.2. Evaluation of antibacterial effects by counting colony-forming units

Figure 4 shows the antibacterial rate (%) determined using quantitative CFU analysis. The relative survival rate of *S. mutans* decreased significantly with the addition of ZnO NPs. In particular, the survival rate was significantly lower in ZnO 4.0 than in CC (*p* < 0.05). The antibacterial rates against *S. mutans* (%) in the CC, ZnO 0.5, ZnO 1.0, ZnO 2.0, and ZnO 4.0 groups were 89.70 ± 3.16, 88.51 ± 2.57, 89.68 ± 1.93, 91.83 ± 2.18, and 97.45 ± 1.37, respectively.

#### 3.2.3. Bacterial Viability

Figure 5 shows representative bacterial viability staining images of *S. mutans* attached to the surfaces. The bacteria in the CC and ZnO 0.5 were generally stained green, indicating that the bacteria were alive. ZnO 1.0 showed live and red-stained dead bacteria. In contrast, visual examination showed that the majority of bacteria in ZnO 2.0 were dead. This finding was more noticeable in ZnO 4.0.

### 3.3. pH Variations

Table 2 presents the data for pH changes in the eluted experimental materials for 1, 2, 4, 7, 14, 21, 30, 60, and 90 days in DW. The pH values of all groups on 1 day were higher than pH 7, and there were no significant differences (*p* > 0.05). After 90 days of immersion, the pH values of all experimental groups containing ZnO NPs were above pH 7, whereas the CC group showed pH values less than pH 7.

### 3.4. Zinc Ion Release

The concentrations of leached zinc ions as a result of the ion release after 1, 2, 4, 7, 14, 21, 30, 60, and 90 d in DW are shown in Figure 6. The results indicated an increasing trend of zinc ion release with an increase in the amount of ZnO NPs (*p* < 0.05). Zinc ion release did not differ significantly between ZnO 1.0 and 2.0 (*p* > 0.05). However, the amount of zinc ions released from ZnO 4.0 was more than 3–4 times higher than that released from the other experimental groups. In contrast, the specimens in the CC group barely released zinc ions.

### 3.5. Depth of Cure

The curing depths of the experimental groups are shown in Figure 7. With an increase in ZnO NP loading, the depth of cure value decreased significantly (*p* < 0.05) in comparison with the control. The depth of cure values in the CC, ZnO 0.5, ZnO 1.0, ZnO 2.0, and ZnO 4.0 groups were 3.26 ± 0.16, 3.12 ± 0.30, 2.90 ± 0.06, 2.80 ± 0.03, and 2.56 ± 0.08 mm, respectively. Nevertheless, the depth of cure values in all groups were greater than 1.5 mm; thus, all experimental groups met the ISO 6874 (2015) requirements.

### 3.6. Water Sorption and Solubility

The water sorption and solubility results are shown in Figure 8. Water sorption and solubility increased as the concentration of ZnO NPs increased (*p* < 0.05). The greatest value for both of these characteristics was found in ZnO 4.0. The water sorption and solubility of CC were significantly lower than those of ZnO 4.0 (*p* < 0.05).

### 3.7. Three−Point Flexural Strength

The three-point flexural strengths of the experimental groups are shown in Figure 9. The three−point flexural strength of CC was higher than that of the other experimental materials. However, the differences among the groups were not significant (*p* > 0.05). The three−point flexural strength values in the CC, ZnO 0.5, ZnO 1.0, ZnO 2.0, and ZnO 4.0 groups were 62.16 ± 1.50, 54.74 ± 3.15, 55.11 ± 2.83, 55.34 ± 6.21, and 54.82 ± 6.36 MPa, respectively.

## 4. Discussion

To the best of our knowledge, this is the first research to comprehensively evaluate the caries-preventing effects of pit and fissure sealants containing ZnO NPs. The combination of conventional pit and fissure sealants with ZnO NPs yielded superior antibacterial properties. In addition, the addition of ZnO NPs had no negative effects on the pH, zinc ion release, depth of cure, water sorption and solubility, and flexural strength.

Oral bacteria can be found in abundance in the marginal space between the tooth and restorative material, and this space is particularly susceptible to dental caries [28]. The acids produced by bacteria in biofilms can cause tooth demineralization [29]. Therefore, inhibition of demineralization and bacterial attachment are critical for preventing secondary caries [30,31]. To address this issue, preventive materials should include antibacterial agents such as ZnO NPs that inhibit secondary caries.

Our evaluations of the antibacterial effect against *S. mutans* through SEM observation, CFU counting, and live/dead staining showed excellent antibacterial effects with 4 wt.% ZnO NPs (Figure 3, Figure 4 and Figure 5). Other studies have shown that zinc acts directly by changing cellular proteins through mechanisms such as indirectly by transmembrane proton translocation by preventing pathogen cells from adhering to surfaces as a result of proteases [32]. By evaluating bacterial attachment (Figure 3), our findings demonstrated that the amount of attachment decreased when the ZnO NP content increased. The live/dead staining approach uses membrane integrity as a stand in for cell viability to discriminate between live and dead bacteria. In addition, live and dead bacteria can also be distinguished on the basis of the relative green and red fluorescence staining of SYTO9 and propidium iodide [33]. As shown in Figure 5, the ZnO 4.0 group showed the reddest fluorescence and almost no green fluorescence, indicating the best antibacterial activity among the experimental groups. Similarly, previous studies using electron microscopy have reported a qualitative reduction in the number of bacterial colonies as the ZnO NPs’ concentration increased [31,34]. These findings are consistent with the quantitative CFU-counting results in the present study (Figure 4).

Table 2 shows that at 90 days, the pH values of the ZnO 2.0 and ZnO 4.0 groups were significantly higher than that of the CC group. Thus, the null hypothesis that the sealants with and without ZnO NPs would show no significant differences in zinc ion release was rejected. The pH of all experimental groups did not show acidity, which may be due to the presence of multiple ions such as F^−^, Na^+^, Sr^2+^, SiO_3_^2−^, BO_3_^3−^, and Al^3+^ in the S-PRG filler. A previous study demonstrated that the pH values of materials containing the S-PRG filler increased rapidly from 4.0 to more than 4.9 within three hours, preventing demineralization of the enamel [35]. Apart from the S-PRG filler, the significant increase in pH after the addition of ZnO NPs may be related to water sorption and solubility. The water sorption and solubility of the experimental materials increased as the ZnO NPs filler content increased. Based on previous findings showing that the water sorption and solubility through the resin matrix affect acid neutralization and ion release in experimental materials [36], the pH results indicate that the higher water sorption and solubility imply greater potential for ion exchange.

As shown in Figure 6, zinc ion release from the experimental materials increased as the ZnO NPs’ filler content increased. Thus, the null hypothesis that sealants with and without ZnO NPs would show no significant difference in zinc ion release was rejected. The increased zinc ion release may have also been responsible for the increasing antibacterial effect with higher ZnO NPs’ filler content: the ZnO 4.0 group, which showed a superior antibacterial effect, also showed the highest level of zinc ion release. More importantly, the ZnO 4.0 group showed zinc ion release of 8 ppm after 90 days, which was approximately 20 times that in the CC group; the continuous release of zinc ions for 90 days may indicate that the antibacterial effect can be maintained for at least 90 days. According to previous studies, the release of zinc ions from polymethyl methacrylate containing 2.5%, 5%, and 7.5% zinc ions differed significantly with increasing zinc ion content. Notably, this ion release was also related to the antifungal effect [37]. In our study, although the ZnO NPs’ content ranged from 0.5 wt.% to 4 wt.%, the findings for the ZnO 0.5 group did not significantly differ from those for the CC group after 90 days. Thus, a ZnO NPs concentration of at least 1 wt.% is required for significant emission in comparison with the CC group. In addition, referring to the results of a previous study, in which approximately 100 times more zinc ions were released in an acidic environment of pH 4.5, this experimental material can be expected to show protective effects by acting as an antibacterial agent in a carious environment [38].

In our measurements of the depth of cure according to ISO 4049 (2019), no significant difference was observed between the CC and ZnO 0.5 groups, while significant differences were observed between the CC group and experimental groups with more than 1 wt.% ZnO NPs. Thus, the null hypothesis that the depth of cure would not differ significantly with and without ZnO NPs was partially rejected. The ability to seal narrow and deep pits and fissures is highly correlated with good polymerization ability [15]. As shown in Figure 7, the depth of cure in the experimental groups significantly decreased with increasing ZnO NP filler content in comparison with that in the CC group. The depth of cure is affected by the curing time, curing unit intensity, amount of light penetration, and aggregate effect of fillers [39]. A previous study showed that ZnO NPs scatter light and obstruct light penetration during curing [40]. Consistent with those findings, the different weight percentages of ZnO NP fillers used in this study may have affected the degree of dispersion in the resin matrix. However, pit and fissure sealants must have a depth of cure greater than 1.5 mm in accordance with ISO 6874 (2015). Our results showed that all experimental groups met these requirements.

In assessments of water sorption and solubility according to ISO 4049 (2019), the CC, ZnO 0.5, and ZnO 1.0 groups showed no significant differences in water sorption. However, water solubility differed significantly in the specimens with and without ZnO NPs. Thus, the null hypothesis that sealants with and without ZnO NPs would show no significant differences in physical properties was partially rejected. In the aquatic environment, the resin matrix and filler composition’s chemical and dimensional stability have the effects on water sorption and solubility [41]. In the current study, ZnO NPs were used as fillers to inhibit secondary caries. According to a previous study, a composite resin containing 2–5 wt.% ZnO NPs does not show adverse effects on water sorption and solubility [42]. As shown in Fig. 6, the water sorption and solubility of the experimental pit and fissure sealants increased as the ZnO NP filler content increased. This may have been due to the ion-release reaction of the ZnO NP filler. Furthermore, a previous study that used hydrated white Portland cement as a filler for pit and fissure sealants found that increasing amounts of filler increased water sorption and solubility because resin composite can allow a calcium and hydroxyl ion release reaction of the filler to occur [36].

Flexural strength measurements using a universal testing machine showed no significant differences among the groups. Thus, the null hypothesis that the sealants with and without ZnO NPs showed no significant differences in mechanical properties was accepted. Because the material can remain in locations that experience mechanical loads during mastication, its mechanical properties are crucial for expanded pit and fissure sealing. A prior study reported that commercially available pit and fissure sealants have a three-point flexural strength of more than 40 MPa [43]. The flexural strength of polymer-based restorative materials should exceed 50 MPa in accordance with ISO 4049 [27]. All average values of the experimental groups in the current study satisfied the ISO requirements, suggesting that pit and fissure sealants containing ZnO NPs may offer useful sites for material placement.

In the present study, ZnO NPs were incorporated into pit and fissure sealants to inhibit dental caries. The findings confirmed that the experimental groups showed higher zinc ion releasing ability than the CC group, resulting in a superior antibacterial effect. The limitations of this study are that we purchased ZnO NPs with a size of fewer than 100 nm and incorporated them into the experimental materials to give an antibacterial effect to the pit and fissure sealant. However, particle size measurement was performed and confirmed to be larger than 100 nm. As a result of observation with the SEM, it was confirmed that rod particles of various sizes were aggregated. Particle size may have affected all experimental results. These need to be demonstrated in further study. In addition, we did not assess the durability of the physical and mechanical characteristics of the materials used for pit and fissure sealants. Thus, additional studies over extended experimental periods will improve our understanding of the long-term effects of these zinc ions releasing pit and fissure sealants in cariogenic oral environments.

## 5. Conclusions

In this study, we prepared a ZnO NP product and incorporated it into commercially available pit and fissure sealants containing S-PRG fillers. To confirm antibacterial properties, morphology observation of attached bacteria, CFU counting, and bacterial viability test were conducted, and it was found that pit and fissure sealants containing 4 wt.% ZnO NPs exhibited significant differences in antibacterial effects. In addition, we discovered that the addition of ZnO NPs had no negative effects on the physicochemical and mechanical properties of the sealants through experiments on pH change, zinc ion release, depth of cure, water sorption and solubility, and flexural strength. Therefore, sealants containing ZnO NPs can be an alternative to conventional pit and fissure sealants and can also be used as a promising material for secondary caries prevention.

## Figures and Tables

**Figure 1 polymers-15-00529-f001:**
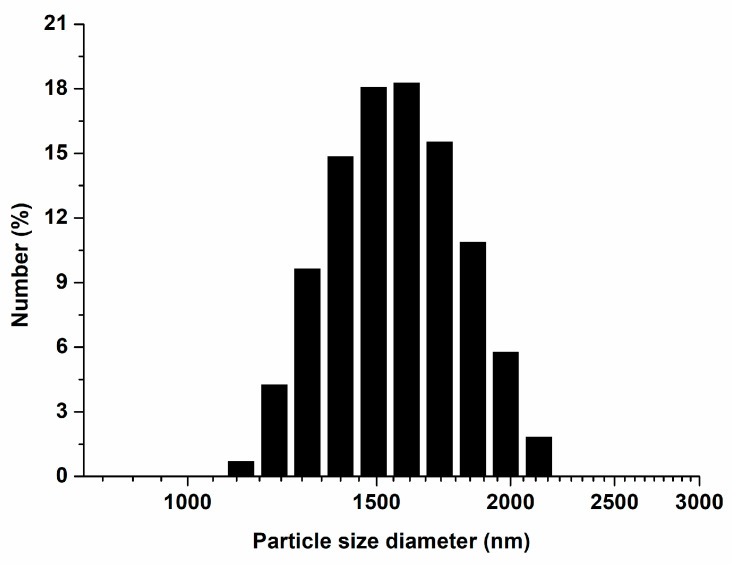
Particle size distribution of ZnO NPs.

**Figure 2 polymers-15-00529-f002:**
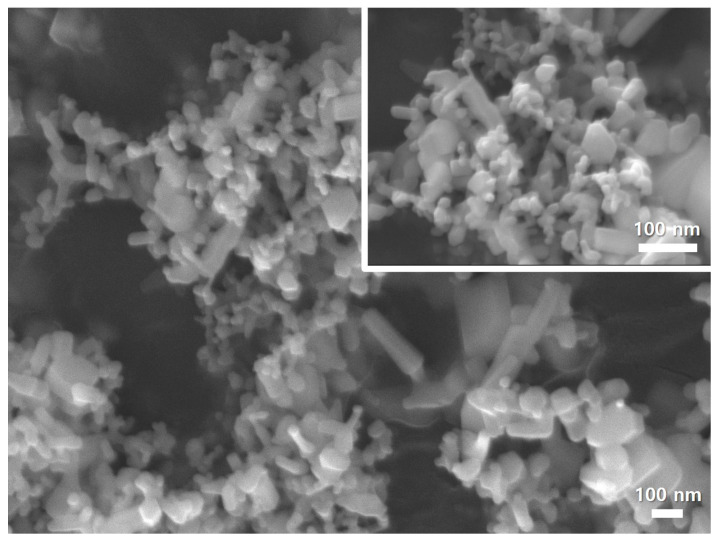
Scanning electron microscopy images of ZnO NPs at magnifications of 100,000× (**top**) and 50,000× (**bottom**).

**Figure 3 polymers-15-00529-f003:**
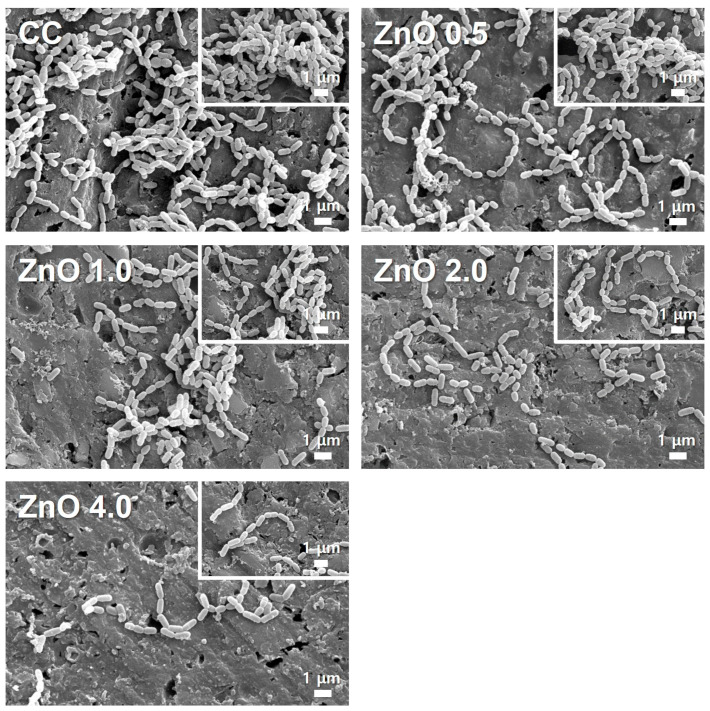
Scanning electron microscopy images of *S. mutans* attached to the surface of the experimental materials with/without ZnO NPs at magnifications of 10,000× (**top**) and 5000× (**bottom**).

**Figure 4 polymers-15-00529-f004:**
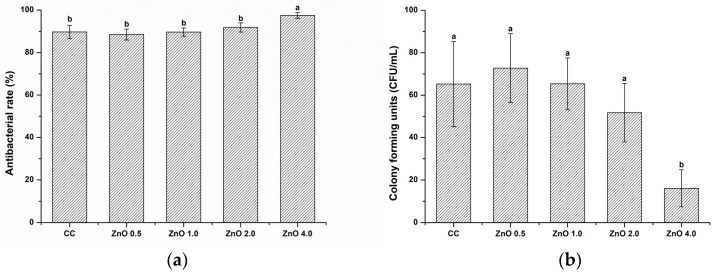
Quantitative evaluations of the antibacterial effect: (**a**) antibacterial rate (%); (**b**) colony-forming units of the experimental materials. Different letters indicate significant differences (*p* < 0.05). The error bars show the standard deviation of the mean values.

**Figure 5 polymers-15-00529-f005:**
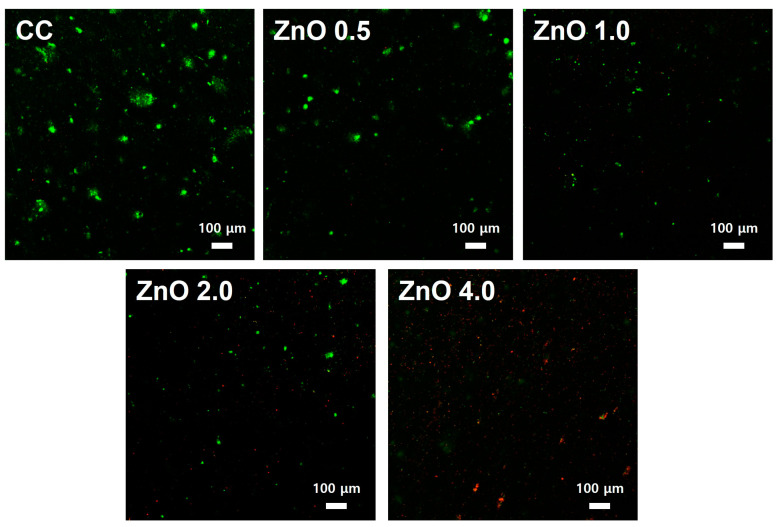
Bacterial viability images of *S. mutans* attached on the experimental material surface with/without ZnO NPs.

**Figure 6 polymers-15-00529-f006:**
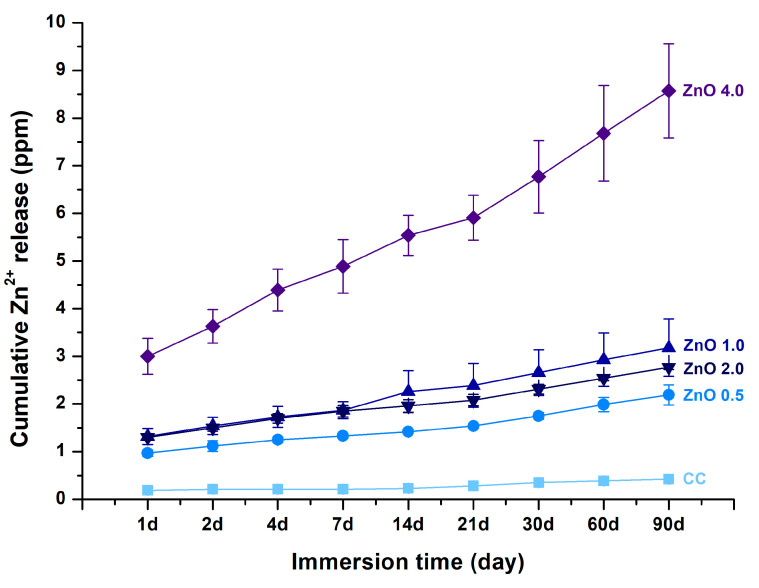
Cumulative Zn^2+^ concentration over 90 days. The error bars indicate the standard deviation of the mean values.

**Figure 7 polymers-15-00529-f007:**
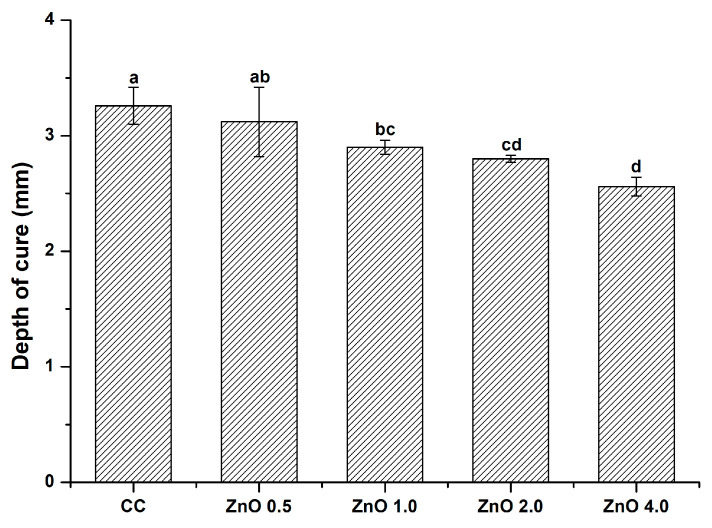
Depth of cure. Different letters indicate significant differences (*p* < 0.05). The error bars show the standard deviation of the mean values.

**Figure 8 polymers-15-00529-f008:**
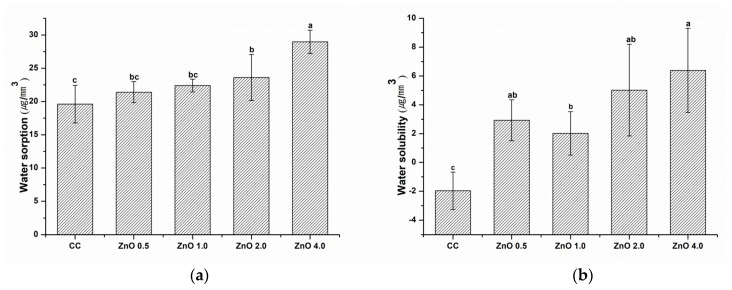
(**a**) Water sorption and (**b**) water solubility. Different letters indicate significant differences (*p* < 0.05). The error bars show the standard deviation of the mean values.

**Figure 9 polymers-15-00529-f009:**
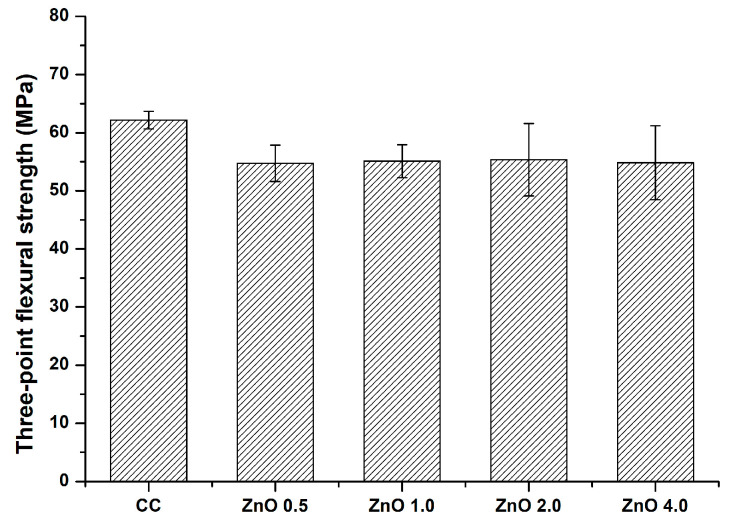
Results of the three-point flexural strength assessments. The error bars show the standard deviation of the mean values. The groups showed no significant differences (*p* > 0.05).

**Table 1 polymers-15-00529-t001:** Compositions of the control and experimental materials in this study (wt.%).

Group Code	Pit and Fissure Sealant	Zinc Oxide
CC	100	0
ZnO 0.5	99.5	0.5
ZnO 1.0	99.0	1.0
ZnO 2.0	98.0	2.0
ZnO 4.0	96.0	4.0

**Table 2 polymers-15-00529-t002:** The pH change for 90 days (means ± SD). The uppercase letters mean comparison by time within the same group.

Group	1 d	2 d	4 d	7 d	14 d	21 d	30 d	60 d	90 d
CC	7.15 ± 0.12 ^Aab^	6.62 ± 0.28 ^CDa^	6.67 ± 0.14 ^Cda^	6.77 ± 0.11 ^BCb^	6.79 ± 0.11 ^BCb^	6.59 ± 0.06 ^Dea^	6.48 ± 0.11 ^Ec^	6.66 ± 0.11 ^Cdab^	6.87 ± 0.08 ^Bd^
ZnO 0.5	7.05 ± 0.03 ^Abb^	6.59 ± 0.20 ^Ca^	6.74 ± 0.09 ^Ca^	6.98 ± 0.09 ^Aba^	6.94 ± 0.05 ^Bab^	6.71 ± 0.07 ^Ca^	6.55 ± 0.07 ^Dbc^	6.59 ± 0.22 ^CDb^	7.09 ± 0.06 ^Abc^
ZnO 1.0	7.04 ± 0.04 ^Ab^	6.60 ± 0.07 ^Dea^	6.64 ± 0.05 ^Dea^	6.84 ± 0.13 ^Bcab^	6.88 ± 0.13 ^Abab^	6.69 ± 0.07 ^Cda^	6.52 ± 0.11 ^Ebc^	6.58 ± 0.19 ^Deb^	7.02 ± 0.08 ^Acd^
ZnO 2.0	7.22 ± 0.19 ^Aa^	6.84 ± 0.11 ^Bca^	6.77 ± 0.10 ^Bca^	6.96 ± 0.10 ^Ba^	6.98 ± 0.14 ^Ba^	6.76 ± 0.09 ^Ca^	6.62 ± 0.04 ^Cab^	6.79 ± 0.15 ^Bcab^	7.22 ± 0.14 ^Aab^
ZnO 4.0	7.13 ± 0.08 ^Bab^	6.72 ± 0.19 ^Da^	6.72 ± 0.16 ^Da^	6.85 ± 0.06 ^Cdab^	7.03 ± 0.11 ^Bca^	6.77 ± 0.17 ^Da^	6.68 ± 0.11 ^Da^	6.89 ± 0.20 ^Cda^	7.37 ± 0.18 ^Aa^

The lowercase letters mean comparison by group within the same time. In addition, different letters indicate that there are significant differences (*p* < 0.05).

## Data Availability

Data are contained within the article.

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
