# Peer review of "Effect of Zinc Oxide Incorporation on the Antibacterial, Physicochemical, and Mechanical Properties of Pit and Fissure Sealants"

_polymers, 2023, doi:10.3390/polym15030529_

Round 1

Reviewer 1 Report

The paper entitled “Effect of zinc oxide incorporation on the antibacterial, physicochemical, and mechanical properties of pit and fissure sealants” described the application of ZnO for pit and fissure sealant. This paper evaluated the antibacterial and mechanical properties of this material. But I think this paper has some flaw which need be revised. The detail comments are as follows:

1.      The authors should clarify other studies of antibacterial pit and fissure sealant. ZnO NPs are well-known antibacterial agents. The authors should state clearly why you choose ZnO NPs.

2.      It is better to characterize the size distribution of ZnO NPs.

3.      Scale bars should be added into Figure 1 and Figure 3.

4.      Can you explain why ZnO-0.5 and ZnO-1.0 had higher colony-forming units?

5.      The curves in Figure 4 are pretty strange. Why ZnO-2.0 released less Zn2+ than ZnO-1.0? The authors must give a reliable explanation.

6.      The mechanical properties of the materials are not enough. Please add more experimental data.

7.      Some references about antibacterial ZnO should be discussed in the introduction part, such as Colloids and Surfaces B: Biointerfaces, 2018, 167, 538; Progress in Organic Coatings, 2021, 151, 106057. ACS Appl. Bio Mater. 2022, 5, 3667–3677.

Reviewer 2 Report

This article presented Effect of zinc oxide incorporation on the antibacterial, physico-chemical, and mechanical properties of pit and fissure sealants. The study is well organized and data is well arranged. The findings would be helpful for future studies. Before recommending this article for publication, there are some shortcomings for that should be resolve.

Briefly discuss in abstract how NPs were prepared or synthesized.

Which characterizations were performed to confirm the synthesis, size and shape.

Line 22-23 should be revise.

Line 62-65 the sentences have many repetition of words. These should be revise.

Line 62. In this paragraph add significance and antibacterial potential of ZnO NPs, biocompatibility and stability in detail by reviewing the literature.

How ZnONPs can be more useful in dental treatments as compared to other material should be added.

Line 64- 65 could be cited with recent relevant study. The following studies could be helpful.

https://doi.org/10.1007/s10534-022-00417-1,

section 2.2.3 could be cited with https://doi.org/10.3390/coatings12101505,

In discussion section the authors should recommend or discuss the possibilities of other types of NPs which may be suitable for dental treatments.

The authors should be consistent while using terminologies or units in the MS such as somewhere the author used ZnONPs, sometimes, ZnO, and sometime ZnO NPs. It should be uniform in the whole MS.

Conclusion is very short. Add future recommendations in the conclusion.

Also check typos and grammatical mistakes in the MS.  

Round 2

Reviewer 1 Report

The authors have addressed my comment. The quality of this manuscript has been improved a lot. I recommend it to publish in this journal.